# Predicting Regions of Local Recurrence in Glioblastomas Using Voxel-Based Radiomic Features of Multiparametric Postoperative MRI

**DOI:** 10.3390/cancers15061894

**Published:** 2023-03-22

**Authors:** Santiago Cepeda, Luigi Tommaso Luppino, Angel Pérez-Núñez, Ole Solheim, Sergio García-García, María Velasco-Casares, Anna Karlberg, Live Eikenes, Rosario Sarabia, Ignacio Arrese, Tomás Zamora, Pedro Gonzalez, Luis Jiménez-Roldán, Samuel Kuttner

**Affiliations:** 1Department of Neurosurgery, Río Hortega University Hospital, 47014 Valladolid, Spain; 2Department of Physics and Technology, UiT The Arctic University of Norway, 9019 Tromsø, Norway; 3Department of Neurosurgery, 12 de Octubre University Hospital (i+12), 28041 Madrid, Spain; 4Department of Surgery, School of Medicine, Complutense University, 28040 Madrid, Spain; 5Instituto de Investigación Sanitaria, 12 de Octubre University Hospital (i+12), 28041 Madrid, Spain; 6Department of Neurosurgery, St. Olavs University Hospital, 7030 Trondheim, Norway; 7Department of Neuromedicine and Movement Science, Norwegian University of Science and Technology, 7034 Trondheim, Norway; 8Department of Radiology, Río Hortega University Hospital, 47012 Valladolid, Spain; 9Department of Circulation and Medical Imaging, Faculty of Medicine and Health Sciences, Norwegian University of Science and Technology (NTNU), 7034 Trondheim, Norway; 10Department of Radiology and Nuclear Medicine, St. Olavs Hospital, Trondheim University Hospital, 7030 Trondheim, Norway; 11Department of Pathology, Río Hortega University Hospital, 47014 Valladolid, Spain; 12The PET Imaging Center, University Hospital of North Norway, 9019 Tromsø, Norway

**Keywords:** glioblastoma, artificial intelligence, MRI, recurrence, radiomics, machine learning

## Abstract

**Simple Summary:**

In this study, we developed a predictive model that employs data from multiparametric structural MRI to predict local recurrence in glioblastoma, providing a practical solution to an issue clinicians face in our daily practice: discriminating edema from tumor infiltration. Predicting the location of these areas at high risk of recurrence will potentially allow for personalizing and optimizing the local treatment of glioblastomas, creating new surgical resection limits and radiotherapy targets. Our findings could potentially improve the survival rate of these patients and open a new line of research that permits a better understanding of the mechanisms of glioma invasion. In addition, we evaluated our results in an external multicenter cohort of patients, thus demonstrating the applicability of the model despite the MRI acquisition protocols and scanner manufacturers. The model will be publicly available through a repository for its implementation by any institution.

**Abstract:**

The globally accepted surgical strategy in glioblastomas is removing the enhancing tumor. However, the peritumoral region harbors infiltration areas responsible for future tumor recurrence. This study aimed to evaluate a predictive model that identifies areas of future recurrence using a voxel-based radiomics analysis of magnetic resonance imaging (MRI) data. This multi-institutional study included a retrospective analysis of patients diagnosed with glioblastoma who underwent surgery with complete resection of the enhancing tumor. Fifty-five patients met the selection criteria. The study sample was split into training (N = 40) and testing (N = 15) datasets. Follow-up MRI was used for ground truth definition, and postoperative structural multiparametric MRI was used to extract voxel-based radiomic features. Deformable coregistration was used to register the MRI sequences for each patient, followed by segmentation of the peritumoral region in the postoperative scan and the enhancing tumor in the follow-up scan. Peritumoral voxels overlapping with enhancing tumor voxels were labeled as recurrence, while non-overlapping voxels were labeled as nonrecurrence. Voxel-based radiomic features were extracted from the peritumoral region. Four machine learning-based classifiers were trained for recurrence prediction. A region-based evaluation approach was used for model evaluation. The Categorical Boosting (CatBoost) classifier obtained the best performance on the testing dataset with an average area under the curve (AUC) of 0.81 ± 0.09 and an accuracy of 0.84 ± 0.06, using region-based evaluation. There was a clear visual correspondence between predicted and actual recurrence regions. We have developed a method that accurately predicts the region of future tumor recurrence in MRI scans of glioblastoma patients. This could enable the adaptation of surgical and radiotherapy treatment to these areas to potentially prolong the survival of these patients.

## 1. Introduction

Glioblastoma is the most common primary neoplasm of the central nervous system, with an overall survival of approximately 15 months from diagnosis in patients who undergo resection and postoperative adjuvant treatment [1]. Despite active research in recent years, including multiple clinical trials, the current therapeutic regimen has not undergone substantial changes in the last decade, and the life expectancy of these patients has not been extended. The current treatment protocol consists of pursuing complete resection of the contrast-enhancing tumor component, followed by adjuvant treatment with chemo- and radiotherapy [2]. Nonetheless, the recurrence and mortality rates inevitably reach 100% in all patients [3].

At the time of diagnosis, glioblastoma is considered widespread because neoplastic cells infiltrate the non-enhancing peritumoral area, as demonstrated by several anatomopathological studies [4,5]. Nevertheless, the non-enhancing peritumor is rarely included as a surgical or radiotherapy treatment target, although it is well-known that more than 80% of recurrences occur near the margins of the resection cavity [6]. The main reason is that, using conventional magnetic resonance imaging (MRI), it is often impossible to visually distinguish a non-enhancing tumor from vasogenic edema, despite specific radiological criteria [7]. The entire peritumoral region has the same contrast on MRI and is shown as a hyperintense signal in T2-weighted (T2w) and fluid-attenuated inversion recovery (FLAIR) images. There is increasing evidence for extending the surgical resection of glioblastoma beyond the margins of the contrast-enhancing region since it could improve patient survival [8]. However, expanding surgical margins is not always feasible because the peritumor can extend to eloquent areas, thus increasing the risk of postoperative neurological deficits. The Response Assessment in Neuro-Oncology (RANO) group recently concluded that less than 5 mL residual non-enhancing tumor volume is prognostically better than complete resection of contrast-enhancing volume alone [9]. However, the non-enhancing tumor varies in size and location, and a more tailored approach toward the non-enhancing tumor burden could be beneficial.

Previous studies have attempted to characterize tumor infiltration into the noncontrast-enhancing region through MRI and in combination with stereotactic biopsies [10] or by applying machine learning and deep learning models [11,12,13,14]. Moreover, several authors have proposed methods to predict regions of future tumor recurrence using MRI-based radiomic features [15,16,17,18,19]. Several of these studies show promising results, but they often require a great variability of image preprocessing, ground truth definitions, feature extraction, and data handling, as well as a need for advanced MRI sequences, which often hinder the generalization and applicability in a clinical setting. Furthermore, most of these studies are based on preoperative MRI, which contains no information about the resection margins, something that is highly important for recurrence predictions. Finally, none of these publications have been validated using external, multi-institutional data.

In the current study, we aimed to develop a method that may not only distinguish between the areas of edema and those of tumor infiltration but also identify sites of possible future tumor recurrence in the peritumor. As a secondary objective, we sought to develop a method that can be used in any institution, regardless of MRI acquisition protocols, without using advanced neuroimaging modalities and usable for both pre- and postoperative scans. This is accomplished using the follow-up MRI for ground truth definition and the postoperative structural multiparametric MRI to extract voxel-based radiomic features as input into a machine learning-based prediction model of glioblastoma local recurrence.

## 2. Materials and Methods

### 2.1. Study Population and Data Description

We conducted a multi-institutional, retrospective analysis of patients who underwent surgery with de novo-diagnosed glioblastoma. The inclusion criteria were as follows: patients with complete resection of the enhancing tumor component, having a preoperative, early postoperative (less than 72 h), and follow-up MRI study, which all included T1-weighted (T1w), T2-weighted (T2w), FLAIR, and T1w contrast-enhanced (T1ce) sequences, as well as diffusion-weighted imaging-derived apparent diffusion coefficient (ADC) maps. The details of the MRI acquisition protocols are shown in Appendix A. In addition, all patients received adjuvant treatment with temozolomide and radiotherapy after surgery according to the Stupp protocol [2]. The exclusion criteria were as follows: pathological diagnoses other than IDH wild-type WHO grade 4 astrocytoma (glioblastoma), partial and subtotal resection, distant recurrences, MRI studies severely distorted due to hemorrhages, postsurgical infarcts or artifacts produced during the acquisition that conditioned images of poor quality. The diagnosis of tumor progression in the follow-up studies was made using the modified RANO criteria [20]. In cases that raised diagnostic doubts, additional follow-up studies were evaluated to discriminate between true tumor recurrence or pseudoprogression. Patients with uncertain progression and cases where the coregistration between the postoperative and follow-up MRIs failed were also excluded.

The initial study population comprised 127 patients from five different institutions (Table 1, column A). Of these, 55 patients met the selection criteria (Table 1, column B). The 40 patients from the Spanish institutions were used for model training. The remaining 15 patients (a Norwegian institution and 2 institutions from the Radiomics Signatures for PrecisiON Diagnostics (ReSPOND) database) [21] were used as the test dataset (Table 1, column C).

### 2.2. Image Preprocessing

The MRI studies were exported from the imaging archive of the respective institutions in the format of Digital Imaging and Communication in Medicine (DICOM) before they were converted to the format of the Neuroimaging Informatics Technology Initiative (NIfTI). The scans for every subject were rigidly registered to the SRI24 anatomical atlas space [22]. Intensity nonuniformity correction was applied as a temporary step to facilitate optimal registration [23].

T1w, T2w, ADC maps, and FLAIR scans were rigidly registered to the transformed T1ce scan for each individual, resulting in coregistered and resampled volumes of 1 × 1 × 1 mm isotropic voxels. The brain was extracted from all coregistered scans using a pre-trained deep learning-based model [24], followed by intensity normalization using Z-scoring. All preprocessing steps were performed using the Cancer Imaging Phenomics Toolkit (CaPTk) [25].

### 2.3. Ground Truth Segmentation

Follow-up and postoperative MRI scans were coregistered using the deformable mode available in CaPTk through the Greedy registration technique [26]. Subsequently, the enhancing tumor was semiautomatically segmented in the T1ce images of the recurrence scan using an edge-based snake evolution method (ITK-SNAP version 3.0 [27]). Likewise, the peritumoral region was segmented in the postoperative MRI using the T1w, T1ce, T2w, and FLAIR scans as input and a hybrid generative-discriminative tumor segmentation method named the boosted glioma image segmentation and registration (GLISTRboost) algorithm [28]. Finally, the overlapping region between the segmented enhancing tumor in the follow-up scan and the peritumor of the postoperative scan was formed. Here, the peritumoral region was manually divided into two subregions to be used as ground truth labels: recurrence, or tumor infiltration, labeled with 1, and nonrecurrence, or edema, labeled with 0, using the intersection and subtraction tools of LifEx version 6.0 [29]. An example of segmentation is shown in Figure 1A,B. The automatic segmentations were reviewed visually, and manual corrections were introduced where the algorithms failed. All segmentations were performed by two neurosurgeons (SC, SG) with more than five years of experience in imaging applied in neuro-oncological surgery. Subsequently, a senior neuroradiologist (MV) with more than 15 years of experience reviewed and adjusted all segmentations of all patients.

### 2.4. Voxel-Based Radiomic Feature Extraction

A total of 4730 voxel-based radiomic features were computed from the peritumoral region of the five multiparametric MRI modalities using the open-source Python package Pyradiomics [30] version 3.0.1. The radiomic features followed the definitions according to the Image biomarker standardization initiative (IBSI) [31] and were divided according to first-order statistical features (19 features) and textural features (75 features). In addition, features were calculated from filtered images using wavelets, Laplacian of Gaussian filters, and local binary patterns. A detailed description of these characteristics and the parameters used in the extraction is provided in Appendix A.

### 2.5. Data Management and Model Training

Inevitably, the number of recurrence voxels was much smaller than the number of peritumor voxels. This class imbalance was handled by randomly undersampling the majority class (nonrecurrence) to match the size of the minority class (recurrence). Four different machine learning-based classifiers were trained: Categorical Boosting (CatBoost), Extreme gradient boosting (XGBoost), Random Forest (RF), and the Light Gradient boosting machine (LightGBM), using Python version 3.9.

### 2.6. Probability Maps and Predicted Recurrence Labels

The output from the machine learning classifiers were voxelwise probabilities of future tumor recurrence, which were represented as a graphical color overlay on the MRI images (Figure 1C). In addition, probabilities were dichotomized into recurrence and nonrecurrence labels, as shown in Figure 1D. The voxels closest to the edge of the surgical cavity had a greater probability of being infiltrated than those located in the most distant regions of the peritumor. Therefore, a correction factor was introduced, which strongly reduced the predicted probability for voxels further than 15 mm from the edge of the segmented surgical cavity. This correction factor was implemented based on previously published anatomopathological findings from [4,5]. Then, the Otsu method [32] automatically determined the threshold used to define the recurrence label. Thus, the probabilities predicted and corrected by the distance factor that exceeded the Otsu threshold were labeled as recurrence.

### 2.7. Model Evaluation

The recurrence prediction performance of the trained model on the external test data was evaluated using two approaches: voxelwise and regionwise. In the first approach, the predicted label of each voxel in the test data was compared to the ground truth using the area under the receiver operating characteristic (ROC) curve (AUC), precision, recall, accuracy, F1 score, and Cohen’s Kappa. However, since it is not a simple segmentation task and due to the biological implications of predicting an infiltrated area subject to evolutionary changes, a second, region-based evaluation approach was developed, taking into account the overall distribution in three-dimensional space. The peritumor was automatically divided into sectors, with the postsurgical cavity as the reference center. The sectors were anterior, posterior, superior, inferior, right, and left, and their angular combinations (Figure 2). This allowed a comparison of whether the predicted recurrent voxels were located within the same sectors as the ground truth recurrence segmentations. Subsequently, the sector predictions were evaluated using the same metrics as the voxel-based approach. Figure 3 shows a schematic representation of the predictive model development process.

### 2.8. Recurrence Prediction in Preoperative MRI Scans

One of the potential implications of obtaining recurrence probability maps of the peritumor region is to adapt the surgical strategy by extending the resection to these areas. The resulting best-trained model was applied to the preoperative MRI scans of the test cohort following the same preprocessing steps mentioned above. The results were qualitatively reviewed by an experienced neurosurgeon (SC).

Once the classifier was trained, the average time required for processing a new patient, including image preprocessing, segmentation, extraction of radiomic features, and model application, was approximately 45 min. A computer with a 2.20 GHz Intel Core i7 processor, 32 GB of RAM, and a 16 GB NVIDIA GeForce RTX 3070 graphics card was used.

## 3. Results

The clinical characteristics of the patients in the training and test cohorts are summarized in Table 2. There were no statistically significant differences between the training and test cohorts in overall and progression-free survival.

Of the 40 patients in the training cohort, the radiomic features were extracted using a total of 1,569,490 voxels, of which 160,366 corresponded to the recurrence label and 1,409,124 to no recurrence label. After random undersampling, a total of 320,732 training voxels were obtained.

Patients from St. Olavs University Hospital had 324,391 test voxels, of which 8475 were recurrence and 315,916 were nonrecurrence. From the ReSPOND database, there were 259,202 test voxels, of which 13,828 were recurrence and 245,374 were nonrecurrence.

The performance metrics of all ML classifiers on the testing dataset are shown in Table 3. The best-performing classification model was the one using the CATboost algorithm. The results of the validation of the model in the external cohort applying the CATBoost algorithm and using sector-based evaluation were as follows: AUC of 0.81 ± 0.09, accuracy of 0.84 ± 0.06, precision of 0.48 ± 0.24, recall of 0.76 ± 0.22, F1 score of 0.53 ± 0.17, and Cohen’s Kappa of 0.45 ± 0.18. Figure 4 shows the individual ROC curves obtained after the evaluation by voxels and sectors. Figure 5 shows the estimated recurrence probability maps in all cases of the test group using the CATBoost algorithm and the follow-up scans where recurrence was diagnosed. Ranking features based on predictive and cumulative importance are shown in Figure 6.

After applying the model in preoperative studies of the testing dataset, recurrence probability maps were obtained in all cases with well-differentiated areas of high infiltration probability and low probability of edema. Despite being unable to make a quantitative evaluation due to the absence of an adequate coregistration system, these predictions revealed a spatial distribution very similar to the sites of future recurrence, proving the existence of an underlying pattern in the image that can be revealed after applying our methodology (Figure 7).

## 4. Discussion

In this retrospective study, we evaluated a machine learning-based approach for predicting tumor recurrence in patients with glioblastoma using radiomic features from postoperative MRI. We found that the predicted recurrence regions were highly correlated with the areas of future recurrence. This suggests that there is a group of features in the multiparametric MRI that reveal a pattern imperceptible to the naked eye. This pattern allows the classifier to distinguish two well-differentiated regions within the peritumoral zone. Therefore, using our methodology, it is possible to predict which areas of the peritumoral region will become tumor-enhancing zones. Thus, the main contribution of our study is the development of a reproducible prediction model with great potential for the application of personalized therapies for this type of neoplasia.

To the best of our knowledge, our study is the first that combines the follow-up MRI to define the ground truth labels and the voxelwise radiomic feature extraction from the peritumoral region of the early postoperative MRI (<72 h). This has the advantage of similar morphology between postoperative and follow-up scans. Early postoperative MRI allows us to define the presence of residual contrast enhancements and quantifies the extent of tumor resection.

Applying the region-based model evaluation, our model achieved mean AUC values of 0.81 and an accuracy of 0.84 in the external testing cohort. These results are superior to those reported by Yan et al. [17] (2020), in which the authors developed a predictive model of recurrence using the voxel-based radiomic features of preoperative MRI (structural, perfusion, and diffusion tensor imaging (DTI)). Similar to our study, the ground truth labels were created through the coregistration of the follow-up scans, but the authors employed preoperative MRI instead of postoperative MRI, as we suggest. The authors reported an overall accuracy in the validation group (n = 20) of 0.78. In the study by Chougule et al. [18] (2021), an accuracy of 0.71 was reported to predict local recurrence in the test group (n = 6). The authors trained a predictive recurrence model using voxel-based radiomic features of the T1ce, FLAIR, and ADC maps. Although retrospective longitudinal data from each subject were collected, only one postoperative scan was used to define the ground truth labels, averaging 143 ± 42 days before recurrence. It is well known that during surgery, some peritumour areas are inadvertently or intentionally resected. Therefore, using preoperative MRI for feature extraction would imply predicting potentially nonexistent regions. Instead, an early postoperative MRI, as suggested by the present work, represents a more robust alternative to predict future tumor recurrence.

The performance of our model also exceeds the results published by Dasgupta et al. [19] (2021). The authors obtained an accuracy of 0.79 and an AUC of 0.61 in the test group (n = 10) using voxel-based radiomic feature extraction from T1ce, T2w, and ADC maps. To train the model, the authors used the features extracted from the peritumoral region of a group of patients with brain metastasis (n = 45) as recurrence labels and the data extracted from the tumor segmentation of a low-grade glioma dataset (n = 36) as recurrence labels. Although this method is innovative, it is not possible to affirm that the composition of low-grade gliomas is the same as that of the non-enhancing tumor areas of glioblastoma.

Our results are slightly lower than those reported by Akbari et al. in 2016, who obtained a mean AUC of 0.84, a sensitivity of 91%, and a specificity of 93% in the testing cohort (n = 34) [16]. The authors used the voxel-based intensity features of the structural, DTI, and dynamic susceptibility contrast-enhanced MRI. Additionally, their model uses preoperative MRI for recurrence prediction, which has limitations, as discussed above.

In the work of Rathore et al. [15] (2018), using previously published methodology [16], the authors included only patients with pathology-proven recurrence diagnoses and added texture features during model training. In the test cohort (n = 31), they obtained an AUC of 0.91 and an accuracy of 0.89, somewhat better than our results. This study used preoperative MRI for ground truth definition, with the limitations discussed earlier. Furthermore, to evaluate predictions, experts manually drew recurrence regions on the preoperative MRI using the follow-up scans as a reference. In contrast, our model does not need expert knowledge to define the true labels since coregistration was used with the follow-up scans.

Only two previous studies mentioned having applied the inclusion criterion that the patients underwent complete resection of the contrast-enhancing tumor [15,16]. This was imperative in our study to ensure that there were no remnants of the enhancing tumor component that could interfere with the analysis.

We acknowledge that the sensitivity of our model is greater than its specificity due to the presence of false positive predicted regions. However, high sensitivity is undoubtedly necessary because we intend to detect a severe event (areas of tumor recurrence) since they are potentially amenable to treatment.

A significant difference in our work compared to previous publications is our method of evaluating predictions. Although the final output of the model is a segmentation integrated by voxels exceeding a certain threshold, predicting areas of recurrence is not an ordinary segmentation task. Two facts with a biological basis must be considered. The first is that the areas of the peritumor are infiltrated, and that will evolve into an enhancing tumor unlikely to have the same size or shape as the enhancing tumor in the follow-up scan. Because these regions undergo evolutionary changes, they naturally tend to grow and invade. The second fact adding disparity between the predicted tumor areas and the ground truth segmentations is that a coregistration of the follow-up and postoperative images has been used for the estimation. As already discussed, this coregistration has limitations, which add uncertainty to the shape and size that the predicted infiltrated peritumor may have. For these reasons, our segmentations cannot be evaluated in a classical way using measures based on overlapping or distance. Instead, we sought to determine whether our predicted recurrence regions were spatially correlated with the site where the tumor would genuinely begin to regrow. By realizing that predicted segmentations need not have high spatial accuracy, we proposed a new regionwise evaluation that brings an intuitive interpretation and evaluation of our predictions. Using this novel approach, we compared whether the regions predicted as recurrence were found in a similar location to the actual site of the enhancing tumor in the follow-up scan.

We highlight that our predictive model has been evaluated in a multi-institutional cohort of patients, which tests the reproducibility of the radiomic features used by the model between different manufacturers of MR scanners and acquisition protocols. Furthermore, our predictive model was trained using the basic or structural sequences of MRI. Since glioblastoma tends to grow predominantly along white matter tracts [33], advanced sequences such as DTI and perfusion MRI could improve our model performance further, as shown in earlier studies [15,16]. However, these sequences are not available in all centers, which would make the model less attractive. In addition, despite the complexity of the methods used here, the final application in a clinical scenario can be carried out with basic computer science and image processing knowledge. Thus, an infiltration probability map of a patient diagnosed with glioblastoma can be obtained in less than 1 h in the DICOM format. It can be easily transferred and used in any neuronavigation or radiotherapy workstation.

We are aware of the limitations of our study. A potential drawback is the lack of anatomopathologic confirmation of regions labeled as recurrences on follow-up scans. As previously mentioned, the diagnosis of recurrence was made according to preestablished radiological criteria. Therefore, we cannot completely rule out the inclusion of a misdiagnosed case of pseudoprogression.

## 5. Conclusions

We have developed and evaluated a model that can predict the location of tumor recurrence from MRI of patients with glioblastoma with high accuracy and sensitivity. Further research focused on the molecular and pathological characteristics of these areas of potential recurrence will allow clinicians to adapt the surgical and radiotherapy treatment to prolong the survival of these patients.

## Figures and Tables

**Figure 1 cancers-15-01894-f001:**
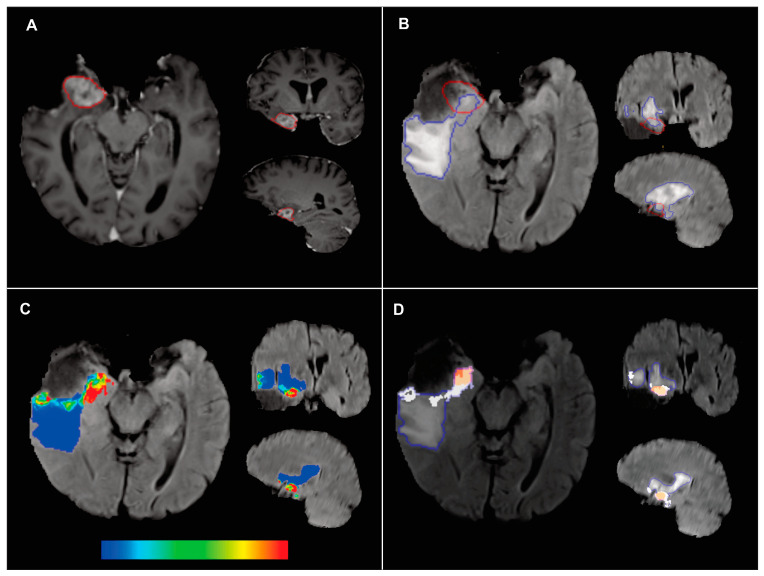
Example of the methodology used for the segmentation task and definition of the ground truth. (**A**) Follow-up MRI of a right temporal glioblastoma recurring in the medial part of the surgical cavity. The segmentation of the newly appearing contrast-enhancing tumor is outlined in red. (**B**) Postoperative MRI FLAIR sequence showing peritumor segmentation in blue and overlay of contrast-enhancing tumor recurrence obtained from the follow-up MRI outlined in red. (**C**) Recurrence probability maps are represented on a color scale ranging from blue (low probability) to red (high probability). A well-defined region was identified in the medial temporal area of the peritumor. (**D**) Predicted recurrence labels are shown in white, the ground truth label in red, and the entire peritumoral region is outlined in blue. The true recurrence zone corresponds well with the predicted labels. Although there are smaller areas of false positives, most nonrecurrence zones have been correctly labeled.

**Figure 2 cancers-15-01894-f002:**
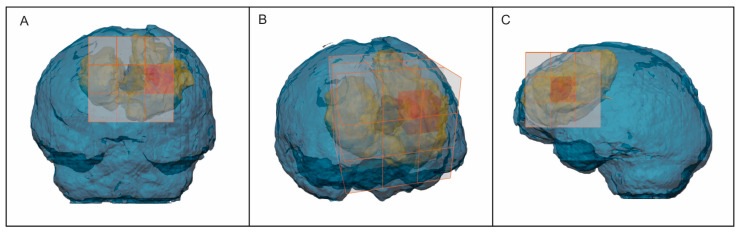
Schematic representation of the region-based model evaluation visualized on frontal (**A**), oblique (**B**), and lateral (**C**) projections. The surgical cavity (black) appears surrounded by the nonrecurrence (yellow) and recurrence (red) regions. The peritumoral region was divided into sectors (transparent cube/orange lines), with the postsurgical cavity as the reference center and the activated sector the recurrence location (red cube).

**Figure 3 cancers-15-01894-f003:**
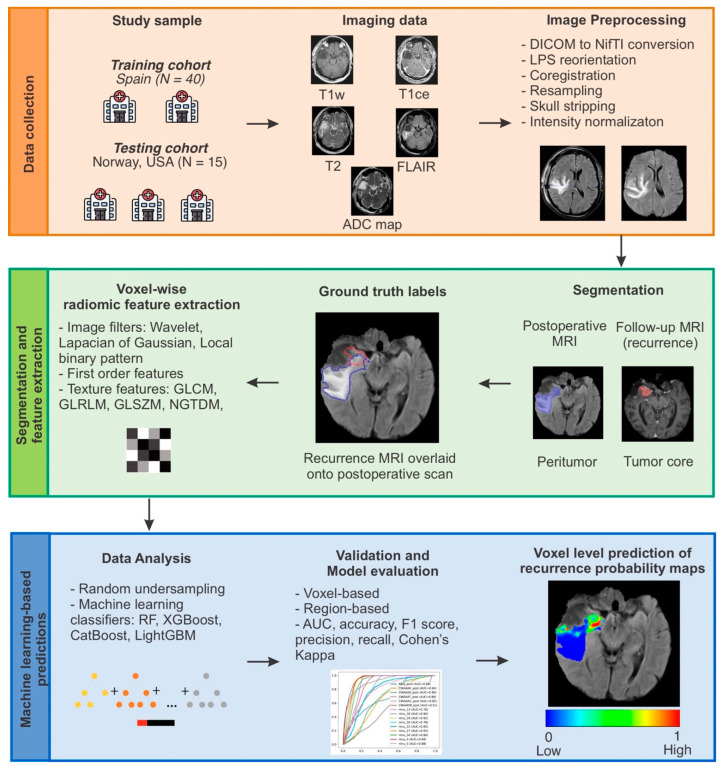
Schematic representation of the analysis workflow.

**Figure 4 cancers-15-01894-f004:**
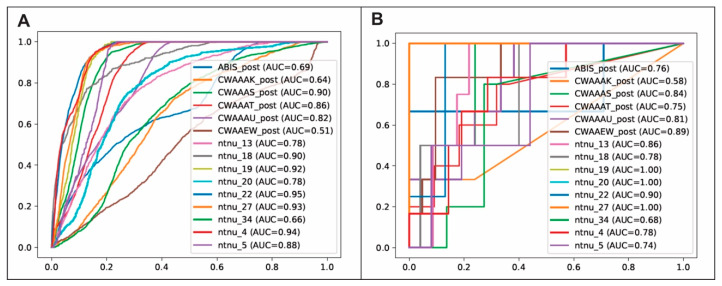
Individual Receiver Operating Characteristic (ROC) curves obtained after applying the evaluation methods based on (**A**) voxels and (**B**) sectors on the external validation cohort and their corresponding values of area under the curve.

**Figure 5 cancers-15-01894-f005:**
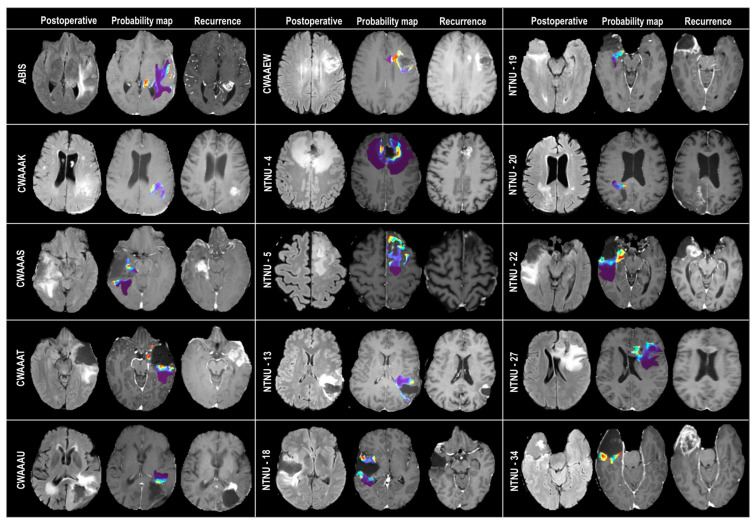
Results of the validation of the model in the external cohort of patients. The columns show the FLAIR sequences of the postoperative MRI, the maps of predicted probabilities of recurrence where red areas correspond to the voxels with the highest probability of becoming an enhancing tumor, and the follow-up T1ce sequences in which the recurrence of the glioblastoma is diagnosed. The rows show the patient IDs.

**Figure 6 cancers-15-01894-f006:**
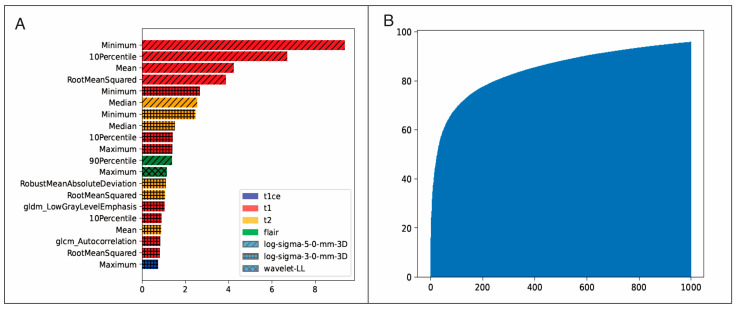
(**A**) CATboost feature importance ranking plot. MRI sequences and image filters are color-coded in the legend. Unless specified in the name, the features are implicitly first-order. (**B**) Cumulative importance versus the number of features.

**Figure 7 cancers-15-01894-f007:**
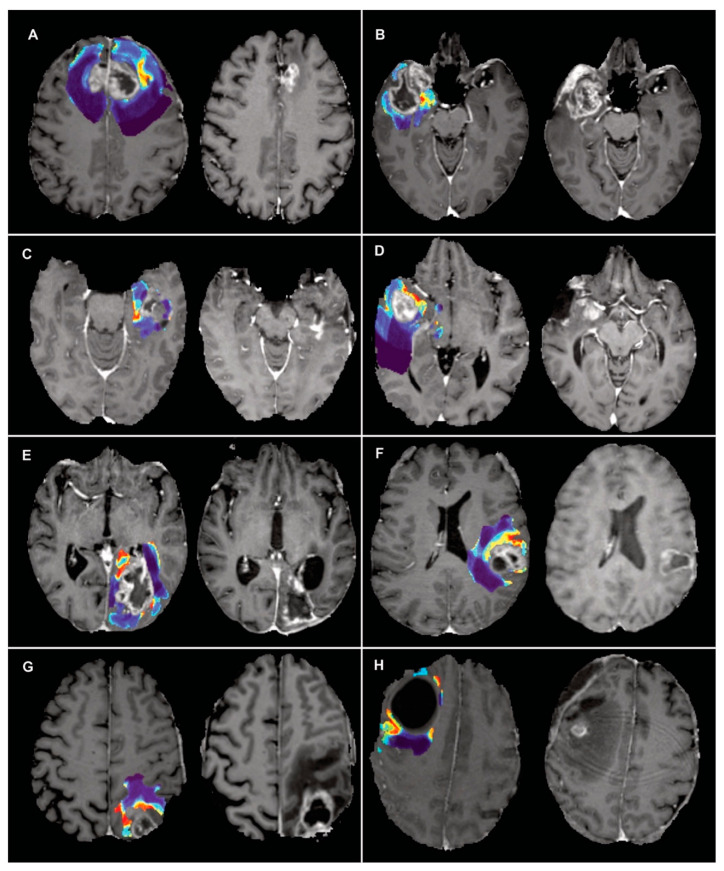
Examples of the application of the model to obtain predictions on preoperative MRI. To the left of each panel, the map of predicted probabilities of recurrence fused with the preoperative T1ce is shown, and to the right the follow-up T1ce where recurrence is diagnosed. (**A**) A bifrontal glioblastoma with an area of a high probability of recurrence located at the left margin of the enhancing tumor. Significant deformation of the brain parenchyma with the collapse of the surgical cavity and recurrence of tumor in the left margin is observed. (**B**) A right temporal glioblastoma with an area of a high probability of recurrence in the non-enhancing medial temporal region. Tumor recurrence enhancement encompasses the medial temporal region and extends into the surgical cavity. (**C**) A left temporal glioblastoma with an area of high risk of recurrence in the posteromedial margin of the enhancing tumor that corresponds to true recurrence in the follow-up study. (**D**) a right temporal glioblastoma with a predicted area of a high probability of recurrence in front of the enhancing tumor that coincides with tumor recurrence located behind the middle cerebral artery. (**E**) A left occipital glioblastoma with a predicted area of recurrence in the ipsilateral hippocampal gyrus. Although the follow-up study shows retraction of the parenchyma and ex vacuo dilation of the occipital ventricular horn, the recurrence of the tumor is located medial to the ventricle at the level of the choroid plexus and internal cerebral veins. (**F**) A left parietal glioblastoma in which the model predicts a zone of recurrence at the anterior and medial margin of the enhancing tumor that is consistent with actual tumor recurrence. (**G**) a left parietal glioblastoma with an area at high risk of recurrence at the anterior margin of the non-enhancing tumor. The follow-up study shows the appearance of enhancing tumor in the anterior margin of the surgical cavity. (**H**) A large right frontal cystic glioblastoma with a predicted area of recurrence at the posterior and lateral margin of the tumor. In the follow-up study, it is observed that despite the deformation and retraction of the parenchyma, an enhancing tumor area has appeared in the posterior and lateral margins of the surgical cavity.

**Table 1 cancers-15-01894-t001:** The number of patients in the study population, study inclusion, and model development.

	A. Study Population	B. Study Inclusion	C. Model Development	
Río Hortega University Hospital, Valladolid, Spain	32	23	40	Training cohort
12 de Octubre University Hospital, Madrid, Spain	28	17		
St. Olavs University Hospital, Trondheim, Norway	35	9	15	Testing cohort
Case Western Reserve University, Cleveland, OH, USA *	20	5		
University of Pennsylvania, Philadelphia, PA, USA *	12	1		
Total	127	55		

* Obtained from the Radiomics Signatures for PrecisiON Diagnostics (ReSPOND) database [21].

**Table 2 cancers-15-01894-t002:** Patient characteristics.

Dataset	Institution	n	Mean Age (SD)	Median Preoperative KPS (IQR)	Median OS (IQR)	Median PFS (IQR)
Training	Río Hortega University Hospital, Valladolid, Spain	23	64 (9)	80 (5)	451 (307)	194 (254)
	12 de Octubre University Hospital, Madrid, Spain	17	56 (13)	80 (10)	466 (217)	186 (203)
Testing	St. Olavs University Hospital, Trondheim, Norway	9	60 (9)	80 (10)	408 (178)	176 (238)
	ReSPOND *	6	NA	NA	447 (271)	262 (251)

SD = standard deviation; KPS = Karnofsky Performance Status; IQR = interquartile range; OS = overall survival in days; PFS = progression-free survival in days; NA: not available. * Obtained from the Radiomics Signatures for PrecisiON Diagnostics (ReSPOND) database [21].

**Table 3 cancers-15-01894-t003:** Performance comparison among machine learning classifiers and model evaluation strategies.

Model Evaluation Strategy	Classifier	AUC	Accuracy	Precision	Recall	F1 Score	Cohen’s Kappa
Voxel-based	RF	0.79 ± 0.13	0.62 ± 0.16	0.15 ± 0.15	0.83 ± 0.16	0.22 ± 0.18	0.12 ± 0.11
	XGBoost	0.80 ± 0.12	0.88 ± 0.12	0.17 ± 0.14	0.19 ± 0.15	0.13 ± 0.07	0.08 ± 0.07
	LightGBM	0.78 ± 0.13	0.87 ± 0.11	0.16 ± 0.16	0.23 ± 0.25	0.13 ± 0.09	0.08 ± 0.07
	CATboost	0.64 ± 0.11	0.84 ± 0.12	0.17 ± 0.13	0.38 ± 0.23	0.18 ± 0.08	0.11 ± 0.07
Region-based	RF	0.85 ± 0.12	0.82 ± 0.09	0.43 ± 0.28	0.75 ± 0.34	0.51 ± 0.28	0.42 ± 0.31
	XGBoost	0.80 ± 0.13	0.81 ± 0.06	0.41 ± 0.22	0.64 ± 0.21	0.46 ± 0.15	0.36 ± 0.16
	LightGBM	0.80 ± 0.11	0.82 ± 0.07	0.45 ± 0.25	0.67 ± 0.23	0.48 ± 0.13	0.38 ± 0.15
	CATboost	0.81 ± 0.09	0.84 ± 0.06	0.48 ± 0.25	0.76 ± 0.22	0.53 ± 0.17	0.45 ± 0.18

Values are expressed as the mean ± standard deviation. AUC = area under the curve; RF = random forest; XGBoost = extreme gradient boosting; LightGBM = light gradient boosting machine; CATboost = categorical boosting.

## Data Availability

Data generated or analyzed during the study are available from the corresponding author by request. All code written in support of this publication is publicly available at: https://github.com/smcch/Predicting_Glioblastoma_Recurrence_MRI (accessed on 19 March 2023).

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
