# Peer review of "Predicting Regions of Local Recurrence in Glioblastomas Using Voxel-Based Radiomic Features of Multiparametric Postoperative MRI"

_cancers, 2023, doi:10.3390/cancers15061894_

Round 1

Reviewer 1 Report

The authors present a predictive model for recurrence of glioblastoma, based on multiparametric MRI. In a retrospective, multi-center study, patients were included if they had undergone complete tumor resection, and if T1w, T2w, contrast-enhanced T1w, FLAIR, and ADC images were available both preoperative, postoperative, and at follow-up. The follow-up images were used as ground truth. Several classifiers were trained, and the best achieved an accuracy of 84%

The manuscript takes a similar approach as several other papers have made during the last few years, with machine learning analysis of contrast-enhanced T1w, T2w, T2-Flair, and ADC/DWI MRI. Overall, this study covers an important research area and represents a reasonable increment in knowledge. The main critique is that the methods are not described with sufficient detail, and that the introduction should be extended with more ackowledgement of previous works, as outlined in the detailed comments below.

Specific points

1.       In several places, the authors claim that their work is novel. I suggest toning down, as several publications exist, which use a very similar approach as the present manuscript. Some works are suggested to cite in the introduction:

a.       Machine Learning-Based Analysis of Magnetic Resonance Radiomics for the Classification of Gliosarcoma and Glioblastoma. Qian Z et al. Front Oncol. 2021 Aug 20;11:699789.

b.       MRI radiomics to differentiate between low grade glioma and glioblastoma peritumoral region. Malik N et al. J Neurooncol. 2021 Nov;155(2):181-191.

c.       Efficient Radiomics-Based Classification of Multi-Parametric MR Images to Identify Volumetric Habitats and Signatures in Glioblastoma: A Machine Learning Approach. Chiu FY, Yen Y. Cancers (Basel). 2022 Mar 14;14(6):1475.

2.       Reference [11] is irrelevant, it is a paper on ultrasound and hepatocellular carcinoma. Instead, use the references above.

3.       The method description lacks detail in several places. In supplementary table 1, please write out also the sequence type (e.g. TSE/FSE, VIBE, TrueFISP, etc). The authors should comment on the long TR-values used on T1w-images in Trondheim and Philadelphia. TR is expected to be 600-800 ms for T1w. Were they possibly T1w-FLAIR?

4.       Line 150: Explain why it was necessary to register to the SRI24 atlas space. Was this done for all time-points (pre, post, follow-up)?

5.       The authors should clearly explain the rationale for using test data acquired with very different MR-scanners and protocols compared to the training data, instead of using cross-validation on the training data.

6.       L277-282: Accuracies should be presented also for the training datasets, not only for testing data.

Author Response

The authors present a predictive model for recurrence of glioblastoma, based on multiparametric MRI. In a retrospective, multi-center study, patients were included if they had undergone complete tumor resection, and if T1w, T2w, contrast-enhanced T1w, FLAIR, and ADC images were available both preoperative, postoperative, and at follow-up. The follow-up images were used as ground truth. Several classifiers were trained, and the best achieved an accuracy of 84%

The manuscript takes a similar approach as several other papers have made during the last few years, with machine learning analysis of contrast-enhanced T1w, T2w, T2-Flair, and ADC/DWI MRI. Overall, this study covers an important research area and represents a reasonable increment in knowledge. The main critique is that the methods are not described with sufficient detail, and that the introduction should be extended with more ackowledgement of previous works, as outlined in the detailed comments below.

Specific points

  1. In several places, the authors claim that their work is novel. I suggest toning down, as several publications exist, which use a very similar approach as the present manuscript. Some works are suggested to cite in the introduction:
  2. Machine Learning-Based Analysis of Magnetic Resonance Radiomics for the Classification of Gliosarcoma and Glioblastoma. Qian Z et al. Front Oncol. 2021 Aug 20;11:699789.
  3. MRI radiomics to differentiate between low grade glioma and glioblastoma peritumoral region. Malik N et al. J Neurooncol. 2021 Nov;155(2):181-191.
  4. Efficient Radiomics-Based Classification of Multi-Parametric MR Images to Identify Volumetric Habitats and Signatures in Glioblastoma: A Machine Learning Approach. Chiu FY, Yen Y. Cancers (Basel). 2022 Mar 14;14(6):1475.

We appreciate your feedback and comments about our work. Following your suggestions, we have added the references mentioned above in the introduction section.

We have focused the introduction on two fundamental issues: First, the need to optimize the local treatment of glioblastomas due to the intrinsic characteristics of this type of neoplasm and the lack of a diagnostic modality to differentiate the areas of edema from those at high risk of recurrence. The second is to acknowledge the existence of previous publications that have pursued this same objective.

We emphasize that there is an essential difference between the simple determination of tumor infiltration versus the prediction of the location of recurrence. Regarding the latter problem, there are relatively few previously published studies, and they all have been included in the references. We acknowledge that there are many publications about applying machine learning techniques in combination with radiomic features in the study of glioblastomas. It is also true that the ML classifiers used here have been previously applied in different medical scenarios. However, the novelty we highlight in our study is how the ground truth was constructed and the source of the radiomic data, in our work, from early postoperative MRI. Moreover, this is the first time that a model for predicting the location of recurrence has used a cohort of patients with strict criteria of total resection.

  1. Reference [11] is irrelevant, it is a paper on ultrasound and hepatocellular carcinoma. Instead, use the references above.

We are very sorry for the error. The correct reference is:

Hu, L.S.; Yoon, H.; Eschbacher, J.M.; Baxter, L.C.; Dueck, A.C.; Nespodzany, A.; Smith, K.A.; Nakaji, P.; Xu, Y.; Wang, L.; et al. Accurate patient-specific machine learning models of glioblastoma invasion using transfer learning. Am. J. Neuroradiol. 2019, 40, 418–425, doi:10.3174/ajnr.A5981

  1. The method description lacks detail in several places. In supplementary table 1, please write out also the sequence type (e.g. TSE/FSE, VIBE, TrueFISP, etc). The authors should comment on the long TR-values used on T1w-images in Trondheim and Philadelphia. TR is expected to be 600-800 ms for T1w. Were they possibly T1w-FLAIR?

We have summarized the methodology to the most relevant aspects that allow its comprehensibility and reproducibility. In addition, we have pursued a balance between the complexity of the information and the potential audience for our work (medical oncologists, radiation oncologists, radiologists, neurosurgeons and data scientists).

Publishing part of the methodology as an online repository will facilitate understanding and application in other centers.

Regarding the repetition times, we have verified that the centers mentioned above used 3D T1-weighted gradient echo sequences.

Following your kind suggestion, we added information about the sequence type and dimension (2D or 3D) to table 1 of the supplementary material.

  1. Line 150: Explain why it was necessary to register to the SRI24 atlas space. Was this done for all time-points (pre, post, follow-up)?

We appreciate your question and the opportunity to clarify this image preprocessing step. The coregistration to the SRI24 atlas was performed at all three-time points (pre, postoperative and follow-up). It is one of the steps required by the software we have used (CaPTK) to apply the segmentation and skull-stripping algorithms described in the methods section.

Also, correcting and normalizing an MRI to a common space, such as the SRI24 template, is important for several reasons in MRI processing:

  • Standardization: By registering all MRIs to a common template, it becomes easier to compare and combine data from different subjects, scanners, and studies.
  • Increased statistical power: When MRIs are registered to a common space, it is possible to perform voxel-wise statistical analyses considering the spatial relationships between voxels across subjects. This increases statistical power and makes detecting subtle differences between groups or conditions easier.
  • Improved accuracy: MRI registration and normalization can improve the accuracy of subsequent analyses by reducing variability and correcting for anatomical differences between subjects. This is particularly important for studies that aim to identify small or subtle changes in brain structure or function.
  • Compatibility with atlases and databases: By registering MRIs to a common template, it becomes possible to use standard brain atlases and databases to label and segment brain regions of interest. This facilitates the interpretation of results and allows researchers to compare their findings with those of other studies.

  1. The authors should clearly explain the rationale for using test data acquired with very different MR-scanners and protocols compared to the training data, instead of using cross-validation on the training data.

Even though there are several publications with a similar objective to ours, the main problem with models based on radiomic features is their limited capacity for generalizability and reproducibility when applied in conditions other than those in which they were trained.

It is well known that the stability of the radiomic features is influenced by the variations between the acquisition protocols of the different centers and the variability between the study subjects.

Therefore, the most critical characteristic that a predictive model for this particular problem must have (predict recurrence sites) must be its ability to be applied in other centers, with images from other scanner manufacturers, acquisition protocols and fields of strength.

Thus, we designed our study using the cohort of Spanish patients as a training group, in which the field of strength and acquisition protocols differ, despite being the same scanner manufacturer (GE). Furthermore, we use an external multi-institutional cohort of patients, representing unseen data, to apply the model and evaluate its performance in conditions closer to a real clinical scenario.

Therefore, we follow the recommendations for developing and evaluating this type of model. In addition, several references 1,2strongly recommend using an external cohort to evaluate the predictions and limiting cross-validation strategies in cases of limited sample size. Finally, we emphasize that our model has been trained voxel-based (320,732 voxels of the training cohort).

  1. Steyerberg EW, Harrell FE. Prediction models need appropriate internal, internal-external, and external validation. J Clin Epidemiol. 2016;69:245-247. doi:10.1016/j.jclinepi.2015.04.005
  2. Cabitza F, Campagner A, Soares F, et al. The importance of being external. methodological insights for the external validation of machine learning models in medicine. Comput Methods Programs Biomed. 2021;208:106288. doi:10.1016/j.cmpb.2021.106288

  1. L277-282: Accuracies should be presented also for the training datasets, not only for testing data.

Since our main goal is to demonstrate how the model works on external data not included in its development, we have included only the results of the model's performance in the test group in the manuscript. Demonstrating that the model works well when applied to patients on whom it has been trained does not reflect any goodness or utility and could be considered conceptually wrong. However, following your kind request, I am sharing the model performance on the training dataset with you.

Reviewer 2 Report

The research article "Predicting regions of local recurrence in glioblastomas using 2 voxel-based radiomic features of multiparametric postoperative 3 MRI" authored by Cepeda et al., is an interesting study. However, there are studies which has already achieved higher mean AUC values what is presented here as 0.81 and an accuracy of 0.84 in the testing cohort.

Author Response

The research article "Predicting regions of local recurrence in glioblastomas using 2 voxel-based radiomic features of multiparametric postoperative 3 MRI" authored by Cepeda et al., is an interesting study. However, there are studies which has already achieved higher mean AUC values what is presented here as 0.81 and an accuracy of 0.84 in the testing cohort.

We appreciate your valuable comments about our research. As you point out, there are previous publications with slightly higher results in accuracy and AUC values. These works have served as a reference for developing our predictive model, and we acknowledge the authors for such a significant contribution. However, it is essential to clarify that the evaluation of our model has been carried out based on voxels/regions and not based on patients. Therefore, the model's inaccuracies should be interpreted considering the methodology used. Furthermore, from a practical point of view, small amounts of misclassified voxels have little influence on the final interpretation of the predictions. We demonstrate this in all the figures included in the manuscript, where it is clearly seen that the model can reliably identify the regions that will become recurrence sites despite isolated misclassified voxels. This fact explains why our model's evaluation results are penalized for trying to find an absolute correspondence between actual and predicted labels at an image resolution of voxels/regions. These considerations are explained in the discussion section.

It is even more important to point out that other studies have used a sample from a single institution, using the same image acquisition parameters that favor the model performance evaluation. On the contrary, we have evaluated the model on unseen external data from three different institutions where other scanner manufacturers and different image acquisition protocols have been used. Still, our results are good enough to ensure the generalizability and reproducibility of our model.

Finally, we highlight that, unlike previous works, we will make the code freely available as a public repository, which will be active from the publication of our work. This will permit other centers to apply the predictive model, thus allowing its reproducibility regardless of the origin of the data. https://github.com/smcch/Predicting_Glioblastoma_Recurrence_MRI

Round 2

Reviewer 1 Report

In general, I am satisfied with the response that the authors have given to my critique. However, I still have issues with the supplementary table 1.

As suggested, the authors have added the sequence type to each sequence in the supplementary table, which is good. However, this has only increased my confusion with the table:

·         For all T2w and FLAIR scans labelled with SE: this just cannot be true. The authors should double-check and label them properly as turbo spin echo (TSE). Possibly, the same may also be the case for the T1ce and T1w scans from Cleveland, please check.

·         The T1w scan from Valladolid: this cannot possibly be a 2D SE. TR/TE are way too short. It must have been a 3D GRE, please check.

·         The T1w scan from Madrid: I do not believe this was a 2D GRE, but instead a 2D SE or 2D TSE, please check.

·         The T1ce and T1w scans from Trondheim and Philadelphia are labelled 3D GRE, which I believe is correct. But then the TR cannot possibly have been 2000 ms or 1760 ms. A TR less than 10 ms is expected. Please double-check.

·         For all scans which actually were GRE (2D or 3D): the flip angle (FA) should be stated, e.g. TR/TE/FA, 7.98 ms/2.57 ms/30°.

It is my firm opinion that the method descriptions should be accurate, sufficiently detailed, and make sense also for radiologists and physicists, who may want to implement the protocol at their own site. Therefore, it is essential that the supplementary table 1 is updated properly.

Author Response

In general, I am satisfied with the response that the authors have given to my critique. However, I still have issues with the supplementary table 1.

As suggested, the authors have added the sequence type to each sequence in the supplementary table, which is good. However, this has only increased my confusion with the table:

Our sincere apologies for the confusion. It has been challenging to trace back and transcribe the metadata from the original DICOMs. We can only access the original images from Valladolid, Madrid and Trondheim. Unfortunately, of the patients in the ReSPOND dataset (Cleveland and Pennsylvania), we only have access to the NifTI files. The consortium coordinators only shared the details of the acquisition protocols already shown in Supplementary Table 1. We acknowledge that there are some missing data (e.g. flip angle of GRE seqs).

  • For all T2w and FLAIR scans labelled with SE: this just cannot be true. The authors should double-check and label them properly as turbo spin echo (TSE). Possibly, the same may also be the case for the T1ce and T1w scans from Cleveland, please check.

We have corrected FSE and TSE depending on the origin of the data. For the Cleveland images, we have added TSE type to the T1w and T1ce sequences after consulting with the information provided by the scanner manufacturer.

  • The T1w scan from Valladolid: this cannot possibly be a 2D SE. TR/TE are way too short. It must have been a 3D GRE, please check.

We have confirmed that it is a 2D FSE type sequence.

  • The T1w scan from Madrid: I do not believe this was a 2D GRE, but instead a 2D SE or 2D TSE, please check.

We have confirmed that it is a 3D FSPGRE type sequence.

  • The T1ce and T1w scans from Trondheim and Philadelphia are labelled 3D GRE, which I believe is correct. But then the TR cannot possibly have been 2000 ms or 1760 ms. A TR less than 10 ms is expected. Please double-check.

As previously mentioned, we have faithfully transcribed the information provided to us by the ReSPOND consortium.

  • For all scans which actually were GRE (2D or 3D): the flip angle (FA) should be stated, e.g. TR/TE/FA, 7.98 ms/2.57 ms/30°.

We have added flip angle information.

It is my firm opinion that the method descriptions should be accurate, sufficiently detailed, and make sense also for radiologists and physicists, who may want to implement the protocol at their own site. Therefore, it is essential that the supplementary table 1 is updated properly.

We appreciate your comments as they will improve our manuscript's quality. However, we must emphasize that although the acquisition protocols' details should be accurately described, this is not a prerequisite for implementing our predictive model in other centers. On the contrary, our results demonstrate that our predictions are reliable regardless of the image acquisition parameters.

Reviewer 2 Report

The research article "Predicting regions of local recurrence in glioblastomas using voxel-based radiomic features of multiparametric postoperative MRI" authored by Cepeda et al., is very good study. The experiments performed by the authors were very elaborate and successful in achieving the specific aims defined in the manuscript. The results were very clear and nicely presented.

Author Response

Thank you very much for your comments.